# Achievement of Both Mechanical Properties and Intrinsic Self-Healing under Body Temperature in Polyurethane Elastomers: A Synthesis Strategy from Waterborne Polymers

**DOI:** 10.3390/polym12040989

**Published:** 2020-04-24

**Authors:** Liangdong Zhang, Teng Qiu, Xiting Sun, Longhai Guo, Lifan He, Jun Ye, Xiaoyu Li

**Affiliations:** 1State Key Laboratory of Organic-Inorganic Composites, Beijing University of Chemical Technology, Beijing 100029, China; zhangld_buct@163.com (L.Z.); qiuteng@mail.buct.edu.cn (T.Q.); xitingsun@163.com (X.S.); 2Beijing Engineering Research Center of Synthesis and Application of Waterborne Polymer, Beijing University of Chemical Technology, Beijing 100029, China; helf@mail.buct.edu.cn (L.H.); yejun@mail.buct.edu.cn (J.Y.); 3Key Laboratory of Carbon Fiber and Functional Polymers, Ministry of Education, Beijing University of Chemical Technology, Beijing 100029, China

**Keywords:** disulfide, self-healing, waterborne polyurethane

## Abstract

Inspired by the growing demand for smart and environmentally friendly polymer materials, we employed 2,2′-disulfanediyldianiline (22DTDA) as a chain extender to synthesize a waterborne polyurethane (WPUR). Due to the ortho-substituted structure of the aromatic disulfide, the urea moieties formed a unique microphase structure in the WPUR, its mechanical strength was enhanced more 180 times relative to that of the material prepared without 22DTDA, and excellent self-healing abilities at body temperature in air or under ultrasound in water were obtained. If the self-healing process was carried out at 37 °C, 50 °C or under ultrasound, the ultimate tensile strength and elongation at break of the healed film could reach 13.8 MPa and 1150%, 15.4 MPa and 1215%, or 16 MPa and 1056%, respectively. Moreover, the WPUR films could be re-healed at the same fracture location over three cutting–healing cycles, and the recovery rates of the tensile strength and elongation at break remained almost constant throughout these cycles.

## 1. Introduction

Inspired by the automatic and autonomous healing of natural organisms [1], research on both extrinsic and intrinsic self-healing materials has recently been attracting increasing interest [2,3]. Extrinsic self-healing is carried out by releasing healing agents from pre-embedded capsules or vascular systems [4,5,6]. Intrinsic self-healing is achieved through reversible chemistry under various stimuli, such as thermal changes [7,8,9], light exposure [10,11], pH changes [12,13], and adding reducing or oxidizing agents [14], as well as through dynamic hydrogen bonds (H-bonds) [15,16,17,18,19], ionic interactions [20,21], metal-ligand coordination [22,23,24,25], and reversible covalent bonds [26,27,28,29,30]. Because they do not necessitate a complex manufacturing process, intrinsic self-healing systems are more suitable for applications such as rubber [31,32], coatings [33], adhesives [34], shape-memory materials [35,36], and electronic skins [37,38,39]. One common polymer widely used in such materials is polyurethane (PUR).

With its high and variable performance, PUR is widely used in aviation engineering, automotive engineering, interior decoration, sports equipment, tissue engineering, and other fields, and it is an essential product in polymer industries [40]. Motivated by the failure problem observed when using PUR, researchers have devoted substantial effort to self-healable PUR materials, hoping that crack propagation could be inhibited by healing microcracks, which would remarkably improve service life as well as safety. By incorporating reversible crosslinks in the backbone or the side chains of PUR, an increasing number of self-healable PURs that heal under various conditions and offer different mechanical properties have been developed to meet application requirements, as summarized in Appendix A. Cheng et al. [41] chose tert-butyl diamine as the chain extender to synthesize room-temperature self-healing PUR with an ultimate tensile strength (σ_0_) of 9.3 MPa. To meet the requirements for e-skins, Bao et al. exploited a dynamic coordinate bond to prepare a self-healing PUR with high toughness and elongation [22,23]. To achieve both a high modulus and glass transition temperature (T_g_), self-healing PURs are generally prepared through Diels–Alder (D–A) bonds [42,43,44]. Visible and ultraviolet light-triggered self-healing PURs have been prepared by introducing a coumarin group [45,46] or a diselenide bond [47], and these materials are potentially applicable as smart coatings.

Since Klumperman et al. first reported a self-healing polymer based on disulfide in 2011 [31,48], dynamic disulfide bonds have attracted increasing attention, especially for self-healing PURs because of their wide adaptabilities, such as tunable healing temperatures from 25 to 100 °C [37,38,49,50,51,52,53,54], multiple stimuli conditions [55,56], and excellent mechanical properties [37,50,51,53,54,56,57,58,59,60]. Zhang et al. [56] employed 2,2′-disulfanediylbis(ethan-1-ol) (HEDS) as a chain extender to prepare a sunlight-driven self-healing PUR with a σ_0_ of 9.5 MPa. Odriozola et al. [49,61] employed 4,4′-disulfanediyldianiline (44DTDA) as a curing agent for isophorone diisocyanate (IPDI) to synthesize room-temperature self-healing PUR. Kim et al. [37] chose 4,4′-dithiodiphenol (44DTDP) as a chain extender and semicrystalline poly(tetramethylene ether glycol) (PTMEG) as a soft segment to synthesize room-temperature self-healing PUR with a σ_0_ of 6.8 MPa. Dong et al. [53] mixed HEDS and 1,4′-butanediol as a complex chain extender to prepare a colorless and transparent moderate-temperature self-healing PUR with a σ_0_ of 25 MPa. Ling et al. [54] reacted high symmetry 4,4′-diphenylmethane diisocyanate with 44DTDP to synthesize high-temperature self-healing PUR with a σ_0_ of 32 MPa.

For the practical applications of self-healing PURs, environmental friendliness should also be considered. Solvent-borne PURs usually cause notable volatile organic compound emission, but these PURs can be replaced by waterborne PUR (WPUR). However, as shown in Scheme 1a, the number of self-healing WPURs reported is far lower than the number of solvent-borne PURs reported, and the healing temperatures of self-healing WPUR are generally moderate. Nevejans [57] employed 44DTDA and bis[4-(3’-hydroxypropoxy)phenyl]disulfide as a chain extender to synthesize high mechanical strength moderate-temperature self-healing WPURs, for which the σ_0_ reached 16.5 MPa. Luo [58] applied HEDS as a chain extender to design a moderate-temperature self-healing WPUR with a σ_0_ of 26 MPa. Chen [60] combined β-cyclodextrin with HEDS to synthesize moderate-temperature self-healing WPUR with a σ_0_ of 19 MPa. To the best of our knowledge, there have been no reports of room-temperature self-healing WPURs with a high mechanical strength (σ_0_ > 10 MPa). Therefore, the design and synthesis of room-temperature self-healing WPURs would be an important addition to the chemistry of self-healing WPURs.

According to Nevejans’ work [57] on the synthesis of moderate-temperature self-healing WPUR from 44DTDA, the T_g_s of the hard phases formed by the reaction of 2,2-bis(hydroxymethyl)propionic acid and 44DTDA with IPDI were in the range of approximately 20 to 60 °C, which would depress the mobility of chain segments containing disulfide bonds, hindering room-temperature self-healing. Recent [62,63,64] molecular simulations of the exchange reaction of aromatic disulfides have indicated that aromatic disulfides with ortho-amine substituents would reduce the H-bonding regularity, which may reduce the healing temperature. Moreover, the asymmetry of the molecular structure around the aromatic disulfide can enhance the self-healing ability [16]. Therefore, we employed 22DTDA, which could form an asymmetric ortho-substituted aromatic disulfide as a chain extender to synthesize self-healing WPUR with a reduced healing temperature, as shown in Scheme 1b.

To avoid excessive creep in the healing process, we chose triethylenetetramine (TETA) to form strong H-bonding interactions with a dissociation temperature of over 100 °C. The incorporation of aromatic disulfides along with the multiple H-bonds and ionic interactions in the PUR network improved both the mechanical strength and its self-healing ability. The optimized WPUR film exhibited an ultimate tensile strength of 18.4 MPa and an elongation at break of 1260%, which was higher than that of the room-temperature self-healing PUR and WPUR, as shown in Scheme 1a. Moreover, ultrasound-induced self-healing and the recycling of the material through hot-pressing were also explored in this work.

## 2. Experimental

### 2.1. Materials

The IPDI (99%) was purchased from TCI Company, Shanghai, China. The 22DTDA (98%), 44DTDA (98%), 4,4′-oxydianiline (44ODA, 98%), 2,2′-Bis(hydroxymethyl)butyric acid (DMBA, 99%), and dibutyltin dilaurate (DBTDA, 97.5%) were purchased from J&K Chemical Company, Beijing, China. The PTMEG (*M*_n_ = 2000 g/mol) was supplied by Shandong Jining Huakai Resin Company, Jining, China. The PTMEG and DMBA were dried in a vacuum at 80 °C before use. The triethylamine (TEA, 99%), TETA (98%), and tetrahydrofuran (THF, 99%) were purchased from Tianjin Fuchen Company, Tianjin, China, and dried with 4 Å molecular sieves before use. The other commercially obtained chemicals were used as received unless otherwise stated. The water used in this work was deionized (DI) water prepared in our laboratory.

### 2.2. Instruments

In this synthesis process, all the glass instruments and Teflon agitators were purchased from Beijing Synthware Glass Company, Beijing, China. The circulating oil bath (HWCL-3) was purchased from Zhengzhou Great Wall Technology Industrial and Trade Company, Zhengzhou, China. The overhead Stirrer (Eurostar 20 digital) was purchased from IKA, Stauffen, Germany. The characterization instruments are in Section 3.

### 2.3. Synthesis

The three-step chain extending process used to synthesize WPU is shown in Scheme 1b, and the formulations used in this work are listed in Table 1. The typical process was as follows: the IPDI (3 g, 13.5 mmol), PTMEG(9.79 g, 4.9 mmol), DMBA (0.74 g, 5 mmol, 5 mas% of solid mass), DBTDA (0.07 g, 0.11 mmol, 0.5 mas% of solid mass), and 10 mL of THF were poured into a 250 mL flask, stirred to homogeneity, and reacted under isothermal conditions of 66 °C, 400 rad/min and nitrogen protection for 4 h. Then, the second chain extension reaction was carried out through the addition of 22DTDA (0.59 g, 2.38 mmol) at 66 °C for 6 h. Then, the flask was cooled to 35 °C. An equimolar amount of TEA (0.5 g, 5 mmol) relative to DMBA was diluted with 10 mL of THF and slowly added to the mixture. Afterward, the DI water (44.35 g, 2.46 mol) was added dropwise into the mixture over 30 min under 1000 rad/min stirring to achieve dispersion of the WPUR. The third chain extension reaction was carried out by the addition of TETA (0.09 g, 0.62 mmol, 1/4 mole relative to the residual NCO) at room temperature in 5 min. Finally, the THF was removed by distillation under reduced pressure and a WPUR dispersion with a solid content of 25 mas% was obtained.

The proper amount of WPUR dispersion was poured into a silicone gel box, dried in an oven at 50 °C for 24 h, transferred into a vacuum oven to completely remove the water, and hot-pressed at 130 °C to form a WPU film with a thickness of approximately 0.5 mm. Hereafter, the synthesized WPUR samples are defined as “P22-X” and “P44-X”, where P22 and P44 stand for the materials prepared using 22DTDA and 44DTDA (Scheme 1c) as the chain extender, and X stands for the mass fraction of 22DTDA or 44DTDA in the WPUR film. The WPUR prepared with a 4 mas% 44ODA (Scheme 1c) as a chain extender is named P44′-4. The WPUR dispersion without the aromatic urea structure is named P-0.

## 3. Characterizations

### 3.1. Dynamic Light Scattering (DLS)

DLS measurements were performed to determine the particle size (*z*-average diameter), distribution (PDI), and zeta potential of the WPUR dispersion by using a Malvern Zetasizer Nano ZS (Malvern Company, Malvern, England) at 25 °C.

### 3.2. Nuclear Magnetic Resonance (NMR)

^1^H NMR spectra were obtained on a Bruker AV-600 spectrometer (Bruker Company, Billerica, MA, USA) using dimethyl sulfoxide-*d*_6_ (DMSO) as the solvent. The chemical shifts (δ, in ppm) in the ^1^H NMR spectra of the synthesized products were used to determine their molecular architecture.

### 3.3. Infrared Spectroscopy

Fourier transform infrared (FTIR) spectra of the samples were collected on a Bruker Tensor 37 spectrophotometer (Bruker Company, Billerica, MA, USA). Scans were acquired in the 4000 to 400 cm^−1^ range, and 16 scans with a resolution of 2 cm^−1^ were acquired for the signal averaging of the IR spectra. The samples were removed from the reaction mixture after different time intervals and spin-coated on the CaF_2_ window.

### 3.4. Gel Permeation Chromatography (GPC)

The number-average molar mass (*M*_n_), the mass-average molar mass (*M*_w_) and the dispersity (*M*_w_/*M*_n_) of the synthesized WPUR materials were determined using a Shimadzu gel permeation chromatography (Shimadzu Company, Kyoto, Japan) system with *N,N’*-dimethylacetamide (DMAC) containing LiCl (0.05 mol/L) as the eluent at 80 °C (flow rate: 1 mL/min). The *M*_n_ and *M*_w_ were evaluated using Shimadzu software and reported as poly(methyl methacrylate) equivalents.

### 3.5. Tensile Tests

Tensile samples were prepared by punching the WPUR films into dumbbell-shaped bars according to GB/T 1040.3–2006. The Young modulus, ultimate tensile strength (σ_0_), and elongation at break (ε_0_) of the original samples were measured using a universal testing machine (MTS systems Company, Eden Prairie, MN, USA) at a crosshead speed of 50 mm/min.

### 3.6. Self-healing and Reprocessing Tests

For the self-healing tests in air, the bars were cut all the way through in the middle to create two pieces. The damaged sample was recombined manually and treated at a preset temperature to examine its self-healing ability. The healing efficiency of the Young modulus (H_E_), ultimate tensile strength (H_σ_) and elongation at break (H_ε_) were calculated by the following formulas:(1)$$H_{\sigma }=\frac{\sigma }{\sigma_{0}}\times 100\%$$
(2)$$H_{\sigma }=\frac{\sigma }{\sigma_{0}}\times 100\%$$
(3)$$H_{\varepsilon }=\frac{\varepsilon }{\varepsilon_{0}}\times 100\%$$
where E, σ, and ε represent the Young’s modulus, ultimate tensile strength, and elongation at break of the healed samples, respectively, and E_0_, σ_0,_ and ε_0_ represent the Young’s modulus, ultimate tensile strength, and elongation at break of the uncut samples at room temperature, respectively.

In the ultrasound-assisted self-healing tests, the sample for tensile strength testing was cut in half using a surgical knife and the pieces were brought back together, spliced manually, and transferred to a water bath at 37 °C. Then, the sample was subjected to ultrasound produced by a Ymnl-450YC cell disruptor (YMNL Company, Nanjing, China) to facilitate the self-healing process. The ultrasonic parameters were 10% of max power and 3000 Hz, and the generator worked intermittently with 20 s intervals after 10 s of work until the cumulative ultrasonic time reached 30 min. After ultrasonication, the healed samples were dried in air for 4 h before testing. As a control, an experimental sample was cut, and the pieces were placed in a water bath at 37 °C for 4 h for self-healing and dried for 4 h in air before testing.

In the recycling and reprocessing tests, the samples were cut into fragments using scissors. The fragments were sandwiched between Teflon plates and hot-pressed at 100 °C, 0.3 MPa for 10 min. The reformed films were punched into dumbbell-shaped bars for tensile testing. The H_σ_ and H_ε_ were also used to evaluate the reprocessability.

### 3.7. Molecular Simulation

The simulations were performed using the Gaussian 09 program package. Two molecular structures were used for these simulations. The structural optimization was simulated at the B3lYP/6-31g (d) level. Vibrational frequency analyses at the same level of theory were performed on all the optimized structures to confirm the stationary points as local minima.

### 3.8. Dynamic Mechanical Analysis (DMA)

A DMA was conducted using a DMA Q800 (TA Company, Boston, MA, USA) operating in tensile mode. The dimensions of the sample were 10 × 4 × 0.5 mm^3^. The storage modulus (E’), loss modulus (E’’) and tanδ were measured at temperatures ranging from −100 to 200 °C, with a heating rate of 3 °C/min under a frequency of 1 Hz and a preload of 0.1 N.

### 3.9. Atomic Force Microscopy (AFM)

The synthesized WPUR dispersion was cast onto clean glass substrates (18 × 18 mm^2^) and heated at 50 °C in an oven to form AFM samples. The surface morphologies of the samples were determined by a tapping-mode AFM (Bruker Company, Billerica, MA, USA), and the data were analyzed using NanoScope Analysis (Bruker Company, Billerica, MA, USA).

### 3.10. Transmission Electron Microscopy (TEM)

Transmission electron microscopy (TEM) images were obtained using a Tecnai G2 F30 (FEI Company, Hillsboro, OR, USA) with an accelerating voltage of 300 kV. The samples were prepared as follows: the synthesized WPUR dispersion was diluted to a solid content of 1 mas%, dropped into a copper mesh, and dried at room temperature in a dustless environment. The diameter of the particles in the TEM image was measured by a Nano Measurer 1.2, and the average particle size (D_T_) and distribution (PDI_T_) were calculated by the following formula:(4)$$D_{T}=\frac{\sum \left(n_{i}\times D_{i}\right)}{\sum n_{i}}$$
(5)$${\mathrm{PDI}}_{T}=\left[\frac{\sum \left(n_{i}\times D_{i}^{4}\right)}{\sum \left(n_{i}\times D_{i}^{3}\right)}\right]/D_{T}$$
where n_i_ and D_i_ represent the number and diameter of the particles, respectively.

### 3.11. Scanning Electron Microscopy (SEM)

The SEM images of the film samples were obtained on a JSM-6700 scanning electron microscope (JEOL Company, Akishima, Japan) with an accelerating voltage of 5 kV. The samples were gold-coated before characterization.

## 4. Results and Discussion

### 4.1. Synthesis of the WPUR Dispersions

The IR spectra recorded before and after the first extension reaction in Figure 1a,b show that the intensity of the band at approximately 1722 cm^−1^ increased due to the C=O groups of urethanes; the intensities of the bands at approximately 3450 and 2260 cm^−1^ decreased in intensity due to the hydroxyl groups and isocyanate groups of IPDI, respectively, suggesting that the IPDI successfully reacted with the PTMEG and DMBA. After adding the 22DTDA monomer, a new band at approximately 1680 cm^−1^ appeared in the spectrum due to the C=O groups of aromatic urea, as shown in Figure 1c, indicating that the 22DTDA reacted with the IPDI immediately after mixing. Shown in Figure 1d, after the second extension reaction, the band at approximately 3380 cm^−1^ due to the aromatic amine groups of 22DTDA disappeared, and the intensity of the band at 2260 cm^−1^ (–NCO) decreased and that of the band at 1680 cm^−1^ increased, indicating that the aromatic disulfide bonds were embedded in the backbone of the PUR chain. After the third chain extension, as shown in Figure 1e, the band at 2260 cm^−1^ disappeared, confirming the complete reaction of the residual isocyanate groups with the TETA. Based on these results in conjunction with the ^1^H NMR and GPC results shown in Appendix A, a WPUR with the anticipated chain structure was synthesized.

The key parameters of WPUR include the particle size (D_z_) and the distribution (PDI) of the dispersion in water, as well as its stability. As shown in Figure 2a, all the WPUR dispersions of the P22 series were stable and showed a more obvious yellow color with the increasing 22DTDA content due to the aromatic disulfide moieties and the increase in particle size. When the 22DTDA content was below 4 mas%, the D_z_ varied only slightly within 5 nm. However, when the 22DTDA content was further increased from 6 to 10 mas%, the D_z_ obviously increased from 35.8 to 95.8 nm but was still within the acceptable range for polymer colloids. Their PDIs increased from 0.084 to 0.268 when the 22DTDA content was increased from 0 to 10 mas%, suggesting a decrease in the dispersibility of the WPUR in water. This trend should have been caused by the hydrophobicity of the aromatic disulfide moieties. The TEM image of P22-4 in Figure 2d shows that the particles are polydisperse, and an average diameter (D_T_) of 33.6 nm and distribution (PDI_T_) of 1.12 were calculated (Appendix A). Moreover, the WPUR prepared with 44DTDA as the chain extender achieved a higher D_z_ (96 nm) and PDI (0.32) than those of the material prepared with 22DTDA as the chain extender, suggesting a worse dispersibility in the P44 series.

The zeta potentials of the particles in the WPUR dispersion are plotted in Figure 2c as a function of the 22DTDA and 44DTDA contents. The negative charges on the particles should have been caused by the ionized carboxylic groups, which are crucial for the stability of WPUR dispersions in water, according to the DLVO theory [65]. We found that all the synthesized WPURs, except P44-4, exhibited an absolute zeta potential of higher than 30 mV, fitting the criteria for moderate stability in colloidal systems. Practically, all the WPURs offered a reasonable storage stability, without the appearance of stratification or sedimentation after storage at room temperature for more than 6 months.

### 4.2. Molecular Simulation

To illustrate the influence of the substitution pattern on the self-healing efficiency, molecular simulations were carried out using the structural models of the aromatic disulfide with ortho- and para-urea substituents (ortho- and para-UADS, respectively). As revealed in Figure 3c,d, the distances between the disulfides were in the ranges of 7.57 to 8.59 Å and from 7.62 to 8.07 Å. When the distance between the S atoms of the aromatic disulfide was in the range of 4.5 to 20 Å, the aromatic disulfides could potentially undergo exchange reactions through chain movement [62]. Thus, the similar distances of the disulfides indicated the similar probabilities of disulfide exchange reactions in the ortho- and para-UADS without other noncovalent interactions. However, the ortho- and para-UADS had different H-bonding structures. The ortho-UADS formed not only two intermolecular H-bonds but also an intramolecular H-bond. The intramolecular H-bond reduced the interactions of the aromatic urea with other urea or urethane groups to depress the *T*_g_ of the blend and hard phases, which could be further confirmed by the DMA results.

### 4.3. Microphase Separation of WPUR Films

The microphase separation of the prepared WPUR films was revealed by the storage modulus (E’) and loss factor (tanδ) measured by the DMA, the results of which are shown in Figure 4; the *T*_g_ of each phase is listed in Appendix A. It is clearly shown in Figure 4a that all the P22 films had an E’ of approximately 3 × 10^3^ MPa at −100 °C. The transition peaks of the tanδ curves at −42 °C (Figure 4b) were a result of the *T*_g_ of the soft phases. Furthermore, the E’ rapidly decreased at temperatures of over 100 °C, and a peak appeared due to the *T*_g_ of the hard phases. It should be noted that a new peak at 11 °C in the tanδ curve could be observed when the 22DTDA content was up to 4 mas%, which suggested the generation of a partially miscible blend phase between the hard and soft phases. All the *T*_g_s of the soft phases remained constant at −42 °C, but those of the blend phases and hard phases increased from 11 to 42.2 °C and from 123 to 140 °C, respectively. Therefore, the blend phase with a low *T*_g_ may have contributed to both mechanical strength and room-temperature healing. The P22-4 film with the highest proportion of blend phase could be determined by comparing the width of the peaks at half maximum. Figure 4c,d show that the *T*_g_s due to the soft phases in P44 are closer to those in P22 with the same disulfide content. However, the *T*_g_s of the blend phases and hard phases in P44 were higher than those in the analogous P22 material, which was consistent with the molecular simulation results.

The influence of the different substitution patterns on the microphase separation could be observed by AFM, and the corresponding phase and height images are shown in Figure 5 and Appendix A, respectively. For P22-4, the hard phases (bright parts) with ball and rod shapes are dispersed in the continuous soft phases (dark parts). For P44-4, double continuous phases with some locally ordered packing structure are clearly visible. The phase contrast in the image of P44-4 is also sharper than that in the image of P22-4. These results are consistent with the DMA results showing that the WPUR film containing para-UADS would form larger and stronger hard phases.

The H-bonding structures in various phases were investigated based on the IR spectra of the P22 films with different 22DTDA contents. Based on previous literature [66], the four IR bands at approximately 1720, 1702, 1675, and 1652 cm^−1^, shown in Appendix A, should have been due to the disordered H-bonds of the carbonyl of the urethane group (disordered UT), the ordered H-bonds of the carbonyl of the urethane group (ordered UT), the disordered H-bonds of the carbonyl of the urea group (disordered UA), and the ordered H-bonds of the carbonyl of the urea group (ordered UA), respectively. The fraction of each H-bond is shown as a function of the 22DTDA content in Figure 6. The amounts of disordered and ordered UA both increased with increasing 22DTDA content, and the contents of disordered and ordered UT decreased. Because the H-bonds within UA were stronger than those within UT, the effects of the H-bonds on the chain mobility should have become more significant. This result, in conjunction with the increasing *T*_g_s of the hard and blend phases in the DMA results, confirmed that the hard and blend phases were mainly composed of ortho-UADS and aliphatic urea.

### 4.4. Self-healing Ability of WPUR Films

#### 4.4.1. Influence of Substitution Pattern on Healing Efficiency

The self-healing processes of P22-4 and P44-4 films are exhibited in Figure 7. The P44-4 film after healing broke again in the marked region as the tensile displacement approached 50 mm (elongation of 150%). However, in sharp contrast, the P22-4 film remained intact even as the tensile displacement approached approximately 250 mm (elongation of 1150%). Moreover, as indicated by their tensile diagrams, shown in Appendix A, Figure 8, and Table 2, the healed P22-4 film had a higher tensile strength than the healed P44-4 film. As we predicted, the aromatic disulfide with ortho-urea substituents (ortho-UADS) reduced the intermolecular H-bonding by forming intramolecular H-bonding, which decreased the *T*_g_s of the blend and hard phases. The decreases in the *T*_g_s of the blend and hard phases resulted in higher healing efficiencies for the P22 films than for the P44 films.

Notably, the healing efficiencies of P44 in our work were lower than those of the WPUR using 44DTDA and linear PURs [49,57,67], which may have been due to the higher values of the E_0_, σ_0_, and ε_0_ of P44 (Table 2) and the more complex H-bonding structure than in the materials reported previously. Without the incorporation of aromatic disulfides, P44′-4 displayed good strength and elongation at break but had no self-healing ability (Table 2). Moreover, based on previous literature [53], linear PUR without disulfide bonds has a mechanical strength similar to that of P22-4 but does not exhibit self-healing. Therefore, the aromatic disulfide moieties were critical in our self-healing process.

The influence of the healing time for P22-4 at different temperatures (50 and 37 °C) was further investigated. The film before healing exhibited a typical elastomer-like stress–strain curve with a Young modulus (E_0_) of 6.5 MPa, an ultimate tensile strength (σ_0_) of 18.4 MPa, and an elongation at break (ε_0_) of 1260%. As shown in Figure 8a,b, good mechanical recovery and high healing efficiency were obtained after healing at both 50 and 37 °C. Even at a healing time of 1 h, the healed P22-4 film showed elastomer-like behavior with good mechanical strength (10.24 and 6.3 MPa at 50 and 37 °C, respectively) and a large deformation (1024% and 684% at 50 and 37 °C, respectively). The healing rate of P22-4 at 50 °C was faster than that at 37 °C, and this could be attributed to the enhanced dynamic reversibility of the aromatic disulfide bonds and H-bonds at higher temperatures. When the healing time was prolonged to 24 h, the σ values after healing at 50 and 37 °C reached 15.4 and 13.8 MPa, and the ε values reached 1215% and 1150%, respectively, which meant that the P22-4 film exhibited a good healing efficiency at room temperature and at moderate temperature.

In Figure 8c,d, notably, the H_E_, H_σ,_ and H_ε_ could not reach 100% even with an extended healing time; instead, they reached equilibrium values. When the healing temperature was 37 °C, the equilibrium values of H_E_, H_σ,_ and H_ε_ approached 81.0%, 75.2%, and 91.4%, respectively, and when the healing temperature was increased to 50 °C, the equilibrium values of H_E_, H_σ,_ and H_ε_ were 90.5%, 83.8%, and 96.9%, respectively.

The equilibrium values of the H_E_, H_σ,_ and H_ε_ of P22-4 after healing at 50 °C were higher than those at 37 °C, which should have been due to the improved chain mobility at a higher temperature. Usually, for phase-separated materials, their modulus and strength are mainly determined by the hard and blend phases, and their elongation is mainly determined by the soft phase. Thus, the higher value of H_ε_ relative to H_E_ and H_σ_ in the healed WPUR indicated that the recovery degree of the soft phase was higher than those of the hard and blend phases in the damaged region. We should also note that the H_E_, H_σ,_ and H_ε_ after healing at both 37 and 50 °C were all lower than 100%, suggesting the incomplete healing of the damaged region. This was due to the high *T*_g_ (over 100 °C) of the hard phase, which caused a depressed chain diffusion between the healed fracture surfaces. A repeated healing test was used to confirm this result.

The repeated healing of P22-4 is verified in Appendix A, and the σ and ε decreased from 15.4 to 5.8 MPa and from 1215% to 880% with the increasing healing cycle number. However, after reprocessing by hot-pressing, as shown in Figure 12e, both the strength and elongation were completely recovered with multiple cycles. The decrease in the mechanical properties with repeated healing cycles was due to the accumulation of unhealed hard phases in the damaged region, and the complete recovery of the mechanical properties after hot-pressing was due to the healing of the hard phases in the damaged region at a high temperature (100 °C).

#### 4.4.2. Influence of the 22DTDA Content on the Healing Efficiency

The stress–strain curves of the P22 films with various 22DTDA contents are displayed in Figure 9, and the values of E_0_, σ_0_ and ε_0_ are summarized in Table 2. Both the modulus and strength of the original P22 films notably increased with the increasing 22DTDA content, which should have been due to the increases in the *T*_g_s of the blend and hard phases. When the 22DTDA content was increased from 0 to 10 mas%, the E_0_ and σ_0_ increased from 0.57 to 24.93 MPa and from 0.16 to 30.0 MPa, respectively. However, the ε_0_ slightly decreased from 1439% to 1011%. The ε_0_ of P-0 was 1205%, which was almost the same as that of P22-4 (1260%).

The tensile curves of P22 after healing at 37 °C are displayed in Figure 9b. The healed P22-2 and P22-4 films exhibited a large plastic deformation, and the healed P22-6 film fractured at a relatively lower plastic deformation. However, the P22-8 and P22-10 films after healing did not exhibit elastic behavior, and the fractures occurred before the clear plastic deformation. When the healing temperature was elevated to 50 °C (Appendix A), the plastic deformations of the healed P22-4 and P22-6 films slightly and obviously increased, respectively. Moreover, the P22-4 and P22-6 films showed elastomer behavior, with plastic deformations of approximately 300%. At the healing temperatures of both 37 and 50 °C, a desirable mechanical strength and effective self-healing could be achieved when the content of 22DTDA was 4 mas%.

To further study the influence of the 22DTDA content on the healing efficiency, the H_σ_ values of films with different 22DTDA contents were measured and are shown in Figure 9c,d. At a healing temperature of 37 °C, we obtained the healing efficiencies (H_σ_ > 75%) of the P-0, P22-2, and P22-4 films. The H_σ_ initially dramatically decreased to 32% as the 22DTDA content approached 6 mas%. When the healing temperature was increased to 50 °C, the H_σ_ values showed a similar trend, and they are all higher than those of the material healed at 37 °C. Even when the healing time was further increased to 3 days, the H_σ_ of the P22 films healed at 37 °C remained almost unchanged, as shown in Figure 9d. Together with the DMA results showing that the *T*_g_s of the blend phases increased from 11 °C to 42 °C with the increasing 22DTDA content from 2 to 10 mas%, we could conclude that the healing efficiency of the P22 films was mainly determined by the *T*_g_ of the blend phases.

#### 4.4.3. Ultrasound-Induced Self-Healing

To illustrate the possibility of self-healing environmentally generated microcracks in our synthesized WPUR, we used ultrasonic cell disruption equipment to produce a vibration shock and damage the P22 films in water. The surface morphologies of the P22 films before and after the ultrasonic disruption were characterized by SEM (Figure 10). The surfaces of P22-2, P22-4, and P22-6 remained complete and they appeared as they did in their original state. However, many small voids were found on the surfaces of P22-8 and P22-10 after the ultrasonic treatment. Such voids were due to the micro defects created by the ultrasonic cavitation, which expanded into micron-sized holes in P22-8 and P22-10. The propagation of the micro defects did not occur in P22-2, P22-4, and P22-6, which could be related to their self-healing of the primary damage. Moreover, the ultrasound-induced acceleration of aromatic disulfide exchange reactions at room temperature has been reported [68], and ultrasonic action may increase the internal temperature of the WPUR materials. Thus, we tried to verify whether ultrasound can promote self-healing, and the healing test results are shown in Figure 11.

As shown in Figure 11a,b, P22-2, P22-4, and P22-6 exhibited decreased strength and elongation after healing in water, which was due to the hindrance of the H_2_O molecules [20]. However, of the three kinds of self-healing conditions, both the strength and elongation achieved after the ultrasound healing were the highest. The σ and ε of P22-6 after the ultrasound healing reached 16 MPa and 1056%, respectively, which were higher than those of P22-6 healed at 50 °C for 24 h in air. These results confirmed that ultrasound could accelerate the healing of P22 films, and we named this phenomenon ultrasound-induced self-healing.

### 4.5. Reprocessability of WPUR Films

Theoretically, linear PUR should perform as a thermoplastic. However, in linear PUR with a high mechanical strength, a high content of strong H-bonding crosslinking makes dissociation difficult even at high temperatures and shear rates [69]. In our case, the thermal dynamic disulfide interactions enhanced the chain mobility, which also enhanced the reprocessability, as shown in Figure 12.

The fragments in Figure 12a, obtained from discarded P22-4 tensile bars, were sandwiched between Teflon plates and hot-pressed at 100 °C under 0.3 MPa for 10 min. During the process, the fragments melted (Figure 12b) and became a homogeneous film (Figure 12c). The mechanical properties of the recycled WPUR films were tested, and the results are compared to those of their original state in Figure 12d. We can see that the reprocessed P22-4 recovers almost 100% of its original properties at a hot-pressing temperature of 100 °C. The H_σ_ and H_ε_ of the P44-4 films after hot-pressing were approximately 60% at a hot-pressing temperature of 100 °C, but these values increased to above 90% when the hot-pressing temperature was increased to 150 °C. However, even when the hot-pressing temperature was 150 °C, the reprocessed P44′-4, without disulfide, showed low mechanical recovery with a H_σ_ of 31.5% and a H_ε_ of 38.6%. These results indicated that the reprocessing ability of the synthesized WPUR materials was largely dependent on the aromatic disulfide moieties in the backbone of the WPUR, and that tuning the H-bonding microenvironments around the aromatic disulfides could effectively reduce the reprocessing temperature. Additionally, the P22-4 sample showed a tensile curve similar to that of the original sample following three cycles of reprocessing (shown in Figure 12e), demonstrating the potential of this material in repeated recycling.

## 5. Conclusions

In this work, we successfully prepared WPUR films with good mechanical properties and a high self-healing efficiency. The optimized film after healing at body temperature could achieve an ultimate tensile strength of 13.8 MPa and an elongation at break of 1150%. In our self-healing WPUR system, the ortho-UADS from 22DTDA was more adaptable and allowed a lower healing temperature than the para-UADS from 44DTDA, because the ortho-UADS easily formed intramolecular H-bonds, which decreased the *T*_g_s of the blend and hard phases. We also noted the importance of the blend phase in improving both the mechanical strength and healing efficiency. Moreover, ultrasound could facilitate self-healing, and films with an ultimate tensile strength of 20.3 MPa could achieve a healing efficiency of 80% in strength after being subjected to ultrasound for only 30 min. In addition, based on the reversible structure around the aromatic disulfide moieties, the film could be recycled and reprocessed under hot-pressing at 100 °C with almost 100% mechanical recovery. Based on the high mechanical properties, body temperature self-healing, and reprocessibility of the WPUR film, it has potential applications in fields such as electronic skin, flexible sensor, intelligent coating, and recyclable adhesive.

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
