# Peer review of "Achievement of Both Mechanical Properties and Intrinsic Self-Healing under Body Temperature in Polyurethane Elastomers: A Synthesis Strategy from Waterborne Polymers"

_polymers, 2020, doi:10.3390/polym12040989_

Round 1

Reviewer 1 Report

Comments to the Author

This work reported  inspired by the growing demand for smart and environmentally friendly polymer materials, the employ 2,2’-dithiodianiline (22DTDA) as a chain extender to synthesize a waterborne polyurethane (WPU). However, the paper have been carefully discussed. Major revision should be done before publication. The following comments are listed for improvement of the paper:  

  1. The references for Scheme 1 are suggested to be removed.
  2. Please confirm the unit used for 1000 rad/s in Line 129, and can this level of speed emulsify WPU?
  3. The format in Section 3 is inconsistent. For example, the bold font shall be applied for “Infrared Spectroscopy. Fourier transform infrared (FTIR)” in Line 163.
  4. Based on the content of Line 225, why does the C = O peak of the urethane group exist in the samples of the mixture of IPDI, PTMEG, and DMBA monomers?
  5. Please clearly specify the details of manufacture information (the manufacturers, the cities and the countries) of all instruments.
  6. Please indicate the potential industrial applications in the Conclusion.
  7. The samples used in this study seem to have contained “dibutyltin dilaurate”. If it is for the purpose of developing environmentally friendly polymer, can a close and similar result be achieved without adding dibutyltin dilaurate?
  8. The presentation format of page numbers of literatures of the References is inconsistent. The repetitive numerical numbers are of some entries are omitted while some are not. Please correct the format after confirming the format requirements of the polymers journal.

Reviewer 2 Report

in a pdf attachment

Reviewer 3 Report

Liangdong Zhang,  Teng Qiu,  Xiting Sun, Longhai Guo, Lifan He, Jun Ye, Xiaoyu Li:  

Achievement of both mechanical properties and intrinsic self-healing under body temperature in polyurethane elastomers: a synthesis strategy from waterborne polymers.

Comments and suggestions for authors:

line 109:

Materials: place (city and/or country) of purchase of chemicals are not always specified.

line 120 and down:

Synthesis part is poorly written. Amounts in moles can improve the synthesis method.

line 125: 400rad/s rotation speed is incredibly (unbelievably) high!

line 133: what is a "silica gel box"?

lines 163, 171, 175: Subtitles should be written in bold.

line 167: calibration method for GPC is missing (what was the used standards?)

The healing process was improved by ultrasonic treatment - this may be explained by local warming.

Round 2

Reviewer 1 Report

The manuscript have been accepted in Polymers.